# Novel HER-2 Targeted Therapies in Breast Cancer

**DOI:** 10.3390/cancers16010087

**Published:** 2023-12-23

**Authors:** Catarina Lopes Fernandes, Diogo J. Silva, Alexandra Mesquita

**Affiliations:** 1Medical Oncology Department, Pedro Hispano Hospital, 4464-513 Matosinhos, Portugal; diogojoao.silva@ulsm.min-saude.pt (D.J.S.); alexandra.mesquita@ulsm.min-saude.pt (A.M.); 2Faculty of Medicine, University of Porto, 4200-319 Porto, Portugal

**Keywords:** breast cancer, HER-2, novel therapies

## Abstract

**Simple Summary:**

HER-2-targeting drugs changed the paradigm of HER-2-positive breast cancer, improving survival rates of an aggressive and lethal disease. Recently, a plethora of new drugs have been approved in different settings, leading to a continuous change in the treatment algorithm of HER-2-positive breast cancer. This current review aims to summarize all the new approved therapeutic options and highlight the new therapeutic options under development.

**Abstract:**

Human epidermal growth factor 2 (HER-2)-positive breast cancer represents 15–20% of all breast cancer subtypes and has an aggressive biological behavior with worse prognosis. The development of HER-2-targeted therapies has changed the disease’s course, having a direct impact on survival rates and quality of life. Drug development of HER-2-targeting therapies is a prolific field, with numerous new therapeutic strategies showing survival benefits and gaining regulatory approval in recent years. Furthermore, the acknowledgement of the survival impact of HER-2-directed therapies on HER-2-low breast cancer has contributed even more to advances in the field. The present review aims to summarize the newly approved therapeutic strategies for HER-2-positive breast cancer and review the new and exploratory HER-2-targeted therapies currently under development.

## 1. Introduction

Breast cancer is estimated to be the type of cancer with the highest incidence in both sexes and the second most lethal all over the world in 2023, with an estimated 300,560 new cases and 43,700 deaths in the United States [1]. The human epidermal growth factor 2 (HER-2) overexpressing subtype accounts for 15–20% of all breast cancer cases [2] and, along with triple-negative, is historically associated with more aggressive behavior and a worse overall prognosis [3,4]. This paradigm has been changing ever since HER-2-targeted therapies were introduced in clinical practice, with excellent clinical responses and increasing survival rates.

HER-2 is one of the four transmembrane tyrosine kinase receptors in the HER family. Its activation leads to the activation of intracellular cell growth pathways, which results in abnormal cell proliferation and apoptosis evasion. Its overexpression in a subset of breast and ovary cancer patients was discovered in the late 1980s [5]. The search for an effective targeted therapy was promptly initiated, leading to the development of trastuzumab in the early 1990s and its later approval to treat HER-2-overexpressing metastatic breast cancer in 1998. Treatment with this targeted monoclonal antibody has been shown to almost double response rates and time to progression in this subset of patients [6]. Hence, ever since the approval of trastuzumab, the investigation of alternative HER-2 inhibitory agents has proliferated, mainly because trastuzumab demonstrated limited efficacy and resistance mechanisms started to come to light [7,8,9].

Regarding this, the search for better treatment options for HER-2-overexpressing breast cancer continues to this day. Current treatment options under investigation include not only HER-2-targeted therapies [monoclonal antibodies, tyrosine kinase inhibitors (TKIs) and antibody-drug conjugates (ADCs)], but also combination therapies with agents from other classes, such as cycline-dependent kinase (CDK) inhibitors, immune-checkpoint inhibitors (ICIs) and cell-therapies, among others. There are currently 662 ongoing interventional studies registered in clinicaltrials.gov for HER-2-positive breast cancer [10], and the number keeps growing.

The present review aims to summarize approved and in-development therapies for HER-2-overexpressing breast cancer and provide a comprehensive perspective on future directions in the treatment of this subset of patients.

## 2. HER-2 Biology and Cellular Mechanisms 

### 2.1. HER-2 Receptor and Signaling Pathway

HER-2 belongs to the family of HER tyrosine kinase transmembrane receptors, which includes HER-1/epidermal growth factor (EGFR), HER-2, HER-3 and HER-4. Whenever activated by a ligand, these receptors stabilize in the cell membrane through homo- or heterodimerization, and hence activate intracellular signaling pathways—mitogen-activated protein kinase (MAPK), phosphatidylinositol 3-kinase (PI3K), AKT and the mammalian target of rapamycin-dependent [11]. These pathways lead to cell division and growth, essential to cancer’s proliferation. The HER family of receptors was found to play an important role in physiological breast tissue growth and differentiation. HER-2 is composed of an extracellular ligand-binding domain (subdivided into four subdomains, I-IV), a transmembrane domain and an intracellular tyrosine kinase domain. Subdomain IV is the one closest to the cell membrane, making it essential to the stabilization of the receptor, while subdomain II is the one structurally more exposed, thus playing an important role in heterodimerization with other HER receptors [12]. HER-2 is the only HER family receptor that does not have a physiological ligand, so it relies on heterodimerization with other HER receptors, particularly HER-3, to activate downstream signaling pathways. The HER-2 receptor is encoded by the HER-2 (also known as *neu* or c-erbB-2) oncogene and was found to be overexpressed in a large number of solid tumors, such as breast, lung, colorectal and gastric cancers [13]. Its overexpression and abnormal ligand-independent activation in cancer cells leads to uncontrolled cell proliferation, evasion of apoptosis and high metastatic potential, thus fulfilling four of the fourteen described hallmarks of cancer [14].

### 2.2. HER-2 Pathological Classification

According to American Society of Clinical Oncology–College of American Pathologists (ASCO-CAP) guidelines, HER-2 positivity is defined as a 3+ score in immuno-histochemistry (IHC) staining in ≥10% of tumor cells or a 2+ score with HER-2 gene amplification through in situ hybridization (ISH) [15].

Recently, the concept of HER-2 low was established, regarding sensitivity to some anti-HER-2 therapies. It is defined as a 2+ score in IHC staining with absence of HER-2 amplification by ISH or simply a 1+ score in IHC, the latter with no need to search for ISH amplification [15].

## 3. Overview of Currently Approved HER-2-Targeted Therapies

Since the initial approval of trastuzumab in 1998, many other HER-2-targeted agents have been developed and some of them have been incorporated into clinical practice, integrating into an increasingly complex therapeutical algorithm.

### 3.1. Monoclonal Antibodies

#### 3.1.1. Trastuzumab

Status: Approved worldwide for use in adjuvant, neoadjuvant and metastatic settings.

Trastuzumab was the first-ever HER-2-targeted monoclonal antibody (mAb) to be developed. It consists of a humanized immunoglobulin (Ig) G1 mAb that inhibits HER-2 by binding to its extracellular subdomain IV. Besides HER-2 signaling pathway inhibition, leading to cell cycle arrest, trastuzumab also induces antibody-dependent cell-mediated cytotoxicity (ADCC) [16,17] and antibody-dependent cell phagocytosis (ADCP) [18], both mediated by trastuzumab Fc-portion’s interaction with Fcγ receptors in the membrane of immune effector cells, such as natural killer (NK) cells and macrophages.

Early studies demonstrated a synergistic effect between trastuzumab and chemotherapy [19], which led to the first clinical trials testing this combination, eventually resulting in trastuzumab approval in combination with chemotherapy in HER-2-overexpressing metastatic breast cancer [6] and later in early breast cancer [20] as well.

Trastuzumab is currently approved (see Table 1):(a)In the metastatic setting, in the first line in combination with docetaxel for 6 cycles and then maintenance with pertuzumab until disease progression, regarding its overall survival benefit when compared with trastuzumab only with docetaxel (56.5 versus 40.8 months, hazard ratio (HR) 0.68) [21], or in monotherapy when the patient has contraindication to taxanes and/or pertuzumab [16];(b)In the neoadjuvant setting in combination with pertuzumab and taxane for locally advanced, inflammatory or early high-risk tumors, regarding its advantage in 5-year progression-free survival (86% versus 81%, HR 0.69) and in 5-year disease-free survival (84% versus 81%), when compared with trastuzumab plus chemotherapy [22];(c)In the adjuvant setting in combination with a taxane for 12 weeks and then as maintenance to complete 17 cycles, which demonstrated a 3-year invasive disease-free survival of 98,7% [23] or in monotherapy for one year after neoadjuvant therapy in patients achieving pathological complete response (pCR), which showed a 5-year disease-free survival of 84% and a 5-year overall survival of 92% [20].

#### 3.1.2. Pertuzumab

Status: Approved worldwide for use in neoadjuvant, adjuvant and metastatic settings.

Pertuzumab is also a humanized mAb but, unlike trastuzumab, it binds the HER-2 subdomain II, preventing HER-2 dimerization with other HER family receptors, specially HER-3 [9]. This precludes HER-2 subsequent mitogenic signaling and thus cell growth and tumor proliferation. Like trastuzumab, pertuzumab also induces ADCC and ADCP, which intensifies its antitumoral activity.

Phase 2 studies showed a good safety profile but limited efficacy with pertuzumab in monotherapy [24]. Pertuzumab and trastuzumab combination therapy was then tested, revealing a favorable safety profile as well as major survival benefits [21].

Pertuzumab is currently approved (see Table 1):(a)In the metastatic setting, in first line in combination with docetaxel for 6 cycles and maintenance with trastuzumab until disease progression [21], as mentioned above;(b)In the neoadjuvant setting, in combination with trastuzumab and docetaxel to treat HER-2-positive early breast cancer with ≥2 cm [22];(c)In the adjuvant setting, in combination with trastuzumab for 1 year after combination with taxane in node-positive or high-risk node-negative disease, in which it demonstrated a 3-year invasive disease-free survival of 92% [25].

There is an ongoing phase 2 clinical trial testing the safety of omitting chemotherapy associated with trastuzumab and pertuzumab in the neoadjuvant setting (NCT01817452). 

#### 3.1.3. Margetuximab

Status: Approved in the United States of America (USA) for use in the metastatic setting.

Margetuximab, previously called MGHA22, is a mAb targeting subdomain IV of the HER-2 receptor, the same as trastuzumab. It has similar receptor affinity, but it differs in its engineered Fc-portion, which allows higher affinity to Fcγ receptor CD16A and lower affinity to inhibitory Fcγ receptor CD32B, leading to increased CD16A-mediated ADCC [26].

When compared to trastuzumab, margetuximab revealed a similar toxicity profile and progression-free survival (PFS) benefit, with no overall survival (OS) benefit in the third line and beyond in a phase 3 study for HER-2-overexpressing metastatic breast cancer [27]. 

Margetuximab is currently approved (see Table 1):(a)In the metastatic setting, in combination with chemotherapy in third line and beyond, showing a 9-month benefit in progression-free survival, although no benefit in overall survival when compared with trastuzumab [27].

An ongoing phase 2 study is testing the replacement of trastuzumab with margetuximab in the neoadjuvant setting, in association with pertuzumab and paclitaxel (NCT04425018).

**Table 1 cancers-16-00087-t001:** Summary of monoclonal antibodies (mAb) currently in use.

mAb	Target	Approved Use	Reference
Trastuzumab	HER-2 subdomain IV	Neoadjuvancy (+pertuzumab +ChT) Adjuvancy (+ChT or monotherapy) Palliative (±pertuzumab ±ChT)	[22] [20,23] [21]
Pertuzumab	HER-2 subdomain II	Neoadjuvancy/Adjuvancy (+trastuzumab ± ChT) Palliative (+trastuzumab ± ChT) in 1L	[22,25] [21]
Margetuximab	HER-2 subdomain IV	Palliative (+ChT) in ≥3L	[27]

ChT: Chemotherapy; 1L: First line; 3L: Third line.

### 3.2. Tyrosine Kinase Inhibitors (TKIs)

#### 3.2.1. Lapatinib

Status: Approved worldwide for use in the metastatic setting.

Lapatinib is a potent and reversible tyrosine kinase inhibitor (TKI) of HER-2 and HER-1/EGFR. It competes with adenosine triphosphate (ATP) in its binding site to the two kinases, due to their similar structure, thereby inhibiting downstream signaling and cell proliferation [28]. Lapatinib also showed efficacy against truncated p95HER-2, which is resistant to trastuzumab extracellular binding and blockade [29].

Lapatinib proved to be effective in the treatment of HER-2-overexpressing metastatic breast cancer, in combination with capecitabine or trastuzumab, after progression with first-line trastuzumab [30]. It also showed benefits in association with endocrine therapy in hormonal receptors (HR)-positive and HER-2-positive breast cancer. Studies in the adjuvant setting showed no significant benefit and increased toxicity when adding or replacing trastuzumab with lapatinib [31,32]. In the neoadjuvant setting, lapatinib showed no benefit in the rate of pathological complete response (pCR) when compared with trastuzumab [33] but it showed a significant benefit when combined with trastuzumab and chemotherapy [34,35]. However, it is not yet approved in this setting.

Besides being active against part of trastuzumab-resistant tumor cells, lapatinib crosses the blood–brain barrier due to its small molecular size and has a significant intracranial antitumoral effect [36].

Lapatinib is currently approved (see Table 2):(a)In the metastastic setting, in combination with capecitabine, which proved a 4-month benefit in time to progression (8.4 versus 4.4 months, HR 0.49) and/or trastuzumab [30].

#### 3.2.2. Neratinib

Status: Approved in the USA and Europe in the adjuvant setting and in the USA in the metastatic setting.

Neratinib, also called HKI-272, has an irreversible and pan-inhibitor effect on HER kinases, acting by covalently binding to a cysteine residue in the ATP binding site, specifically, residues Cys-773 in HER-2 and Cys-805 in HER-1 [37,38]. It also leads to the dissociation and degradation of HSP90.

Neratinib did not show any benefit in the metastatic setting in HER-2-overexpressing breast cancer when compared to lapatinib and capecitabine [39,40], but it was revealed to be effective when in combination with capecitabine [41], warranting its approval in the metastatic setting beyond third line. In the early-disease setting, neratinib showed a benefit in recurrence-free survival when used for one year in extended adjuvant therapy after 2-year neoadjuvant/adjuvant trastuzumab therapy, specifically in HR-positive patients [42]. In the neoadjuvant setting, neratinib showed benefits when compared to trastuzumab [43] and when added to it [44].

Neratinib is currently approved (see Table 2): (a)In the metastatic setting, in combination with capecitabine, in third line and beyond, showing a 2.2-month benefit in progression free-survival (8.8 versus 6.6 months, HR 0.76) [41].(b)In the adjuvant setting, for 1 year, after completing 1 year of adjuvant trastuzumab, regarding its 5-year invasive disease-free survival benefit of 2.5% (90.2% versus 87.7%, HR 0.73) [42].

Three phase 2 trials are currently evaluating neratinib in the neoadjuvant setting (NCT01042379; NCT04886531; NCT05919108) and one in the adjuvant setting in patients without pCR after neoadjuvant treatment (NCT05388149). 

#### 3.2.3. Tucatinib

Status: Approved worldwide for use in the metastatic setting.

Tucatinib is a reversible HER-2-selective TKI, with very little EGFR inhibition, which guarantees a more favorable tolerability profile than other TKIs [45]. Its exact mechanism of action is not yet clarified, but a molecular study points to a strong and stable binding to the ATP pocket in HER-2 tyrosine kinase [46]. Tucatinib demonstrated greater efficacy, in combination with capecitabine and trastuzumab, in third-line treatment of HER-2-overexpressing metastatic breast cancer with or without brain metastases [47]. 

Tucatinib is currently approved (see Table 2):(a)In the metastatic setting, beyond third-line treatment, in combination with capecitabine and trastuzumab. This combination proved to increase progression-free survival by 2.7 months (7.6 versus 4.9 months, HR 0.57) and overall survival by 5.5 months (24.7 versus 19.2 months, HR 0.73) when compared with trastuzumab plus capecitabine [47].

A phase 3 trial, HER2CLIMB-05, is currently evaluating the efficacy of tucatinib associated with trastuzumab and pertuzumab in patients not previously treated with a TKI (NCT05132582). There are four other phase 2 studies evaluating combinations of tucatinib, trastuzumab and different chemotherapy agents in the palliative setting (NCT05748834; NCT05458674; NCT05583110; NCT05955170). The previously mentioned phase 2 basket trial in the neoadjuvant setting also has one arm evaluating the efficacy of neoadjuvant tucatinib combined with trastuzumab and pertuzumab in HER-2-positive breast cancer patients (NCT01042379). 

#### 3.2.4. Pyrotinib

Status: Approved in China for use in the metastatic setting.

Pyrotinib is the most recently developed and approved TKI for HER-2-overexpressing metastatic breast cancer. It is an irreversible pan-HER TKI that covalently binds to the catalytic region of the HER-2 kinase [48]. It has proved, so far, to be effective in the metastatic setting in combination with capecitabine when compared to placebo or lapatinib in Chinese patients [49,50]. Pyrotinib is not yet approved for marketing in the European Union nor in the United States of America as a result of the lack of data on efficacy, tolerability and cost-effectiveness in a broader population [51].

Pyrotinib is currently approved (see Table 2):(a)In the metastatic setting in combination with capecitabine (only in China), showing a significantly higher progression-free survival (18.1 versus 7.0 months, HR 0.36) and higher overall response rate (78.5% versus 57.1%) when compared with lapatinib and capecitabine [49,50].

There are multiple ongoing clinical trials involving pyrotinib, mostly held in China. There are five studies in the metastatic setting (NCT05346861, NCT05255523, NCT04605575, NCT05429294, NCT04246502), with one phase 3 study and two studies in first-line treatment. There are six ongoing studies in the adjuvant setting (NCT04254263, NCT05841381, NCT05861271, NCT04659499, NCT05880927, NCT05834764), two of them phase 3, and seven studies in the neoadjuvant setting (NCT04929548, NCT05659056, NCT06000917, NCT04917900, NCT05430347, NCT04900311, NCT04290793).

**Table 2 cancers-16-00087-t002:** Summary of tyrosine kinase inhibitors approved in clinical practice.

TKI	Target	Effect	Approved Use	Reference
Lapatinib	HER-1 and 2	reversible	Palliative (+capecitabine and/or trastuzumab)	[30]
Neratinib	HER-1, 2 and 4	irreversible	Adjuvant (after 1y trastuzumab ±pertuzumab) Palliative (+capecitabine)	[42] [41]
Tucatinib	HER-2	irreversible	Palliative (+capecitabine +trastuzumab)	[47]
Pyrotinib	HER-1, 2 and 4	irreversible	Palliative (+capecitabine)	[49,50]

### 3.3. Antibody-Drug Conjugates (ADCs)

#### 3.3.1. Trastuzumab Emtansine

Status: Approved worldwide for use in the metastatic and adjuvant settings.

Trastuzumab emtansine (T-DM1) is the first-ever approved antibody–drug conjugate for solid tumors, approved for use in HER-2-positive metastatic breast cancer since 2013 [52] and in the adjuvant setting since 2019 [53]. It is composed of the IgG1 mAb trastuzumab connected to emtansine, a cytotoxic agent that acts by microtubule inhibition, through a non-cleavable linker [54]. 

Studies demonstrated T-DM1’s superiority in terms of rate of response, survival and toxicity profile, when compared with the previous standard second-line treatment, lapatinib and capecitabine, leading to its early approval in this setting [52]. Later studies also showed its efficacy in the adjuvant setting [53].

T-DM1 is currently approved (see Table 3):(a)In the metastatic setting, as second-line treatment for HER-2-overexpressing metastatic breast cancer after trastuzumab therapy, as the EMILIA trial showed a 3.2-month increase in progression-free survival (9.6 versus 6.4 months, HR 0.65) [52] and a 4-month increase in overall survival (29.9 versus 25.9 months, HR 0.75) [55] when compared with lapatinib plus capecitabine;(b)In the adjuvant setting, for residual disease after neoadjuvant chemotherapy combined with trastuzumab and pertuzumab, showing an increase in invasive disease free survival (88.3% versus 77.0% at 3 years, HR 0.50) when compared with trastuzumab [53].

A recent and still unpublished phase 2 trial, ATEMPT, evaluated the efficacy of adjuvant T-DM1 in stage I disease, compared with paclitaxel and trastuzumab (NCT01853748).

#### 3.3.2. Trastuzumab Deruxtecan

Status: Approved in the USA and Europe for use in the metastatic setting.

Trastuzumab deruxtecan (T-DXd) is composed of the mAb trastuzumab connected to deruxtecan, a topoisomerase I inhibitor that causes cell apoptosis and double-stranded DNA breaks. These two compounds are connected by a linker that is cleaved by cathepsins, leading to its selective cleavage inside tumor cells, where cathepsins are upregulated. The cleavability of the ADC also allows a bystander killer effect, a mechanism that expands the drug’s cytotoxic effect to surrounding tumor cells. It also differs from T-DM1 in its higher antibody:drug ratio of 1:8 (T-DM1 has a 1:3.5) [56].

T-DXd proved to be effective in monotherapy in the treatment of HER-2-overexpressing refractory metastatic breast cancer [57,58], even when compared with T-DM1 [59]. More recent studies demonstrated that T-DXd is also effective in the treatment of HER-2-low metastatic breast cancer [60].

T-DXd is currently approved (see Table 3):(a)In the metastatic setting in HER-2-positive breast cancer, in the second line or further, regarding its benefit in delaying disease progression (75.8% of patients had no disease progression at 12 months versus 34.1% with T-DM1, HR 0.28) and delaying death (94.1% were alive at 12 months compared with 85.9%, HR 0.55) [59];(b)In the metastatic setting in HER-2-low breast cancer, regarding its 4.7-month benefit in progression-free survival when compared with physician’s choice chemotherapy (HR 0.51) and its 6.4-month benefit in overall survival (HR 0.64) [60].

There are currently two ongoing trials (phase 2 and 3) evaluating T-DXd as a first-line treatment in metastatic HER-2-positive breast cancer and one study evaluating it as a first-line treatment in HER-2-low patients (see Table 4). A phase 3 study is testing T-DXd as an adjuvant therapy after neoadjuvant treatment with residual disease. Three other studies, one phase 3, are evaluating the role of T-DXd in the neoadjuvant setting (see Table 4).

**Table 3 cancers-16-00087-t003:** Summary of antibody–drug conjugates approved in clinical practice.

ADC	Cytotoxic	Linker	Drug-to-Antibody Ratio	Approved Use	Reference
T-DM1	Emtansine (DM1)	Non-cleavable	3.5:1	Adjuvant HER-2+ EBC HER-2+ MBC ≥ 2L	[53] [52]
T-DXd	Deruxtecan (DXd)	Cleavable	8:1	HER-2+ MBC ≥ 2L HER-2-low MBC	[58,59] [60]

EBC: Early breast cancer; MBC: Metastatic breast cancer; 2L: Second line; 3L: Third line.

**Table 4 cancers-16-00087-t004:** Summary of ongoing clinical trials involving HER-2-targeted agents already in use in clinical practice.

Anti-HER-2 Agent	Clinical Trials (Phase)	Population	Treatment in Study	Status
Trastuzumab	-	-	-	-
Pertuzumab	NCT01817452 (II)	HER-2+ EBC	Neoadjuvant P+T ±ChT	Recruiting
Margetuximab	NCT04262804 (II) NCT04425018 (II)	HER-2+ MBC ≥ 3L HER-2+ EBC	Margetuximab +ChT Neoadj margetuximab +P +ChT	Completed; unpublished Recruiting
Lapatinib	-	-	-	-
Neratinib	NCT01042379 (II) NCT04886531 (II) NCT05919108 (II) NCT05388149 (II)	HER-2+ EBC HER-2+ EBC HER-2m EBC HER-2+ EBC	Neoadj neratinib Neoadj neratinib +T +ET Neoadj neratinib Adjuvant neratinib + T-DM1	Recruiting Recruiting Not yet recruiting Recruiting
Tucatinib	NCT05132582 (III) NCT05748834 (II) NCT05458674 (II) NCT05583110 (II) NCT05955170 (II) NCT01042379 (II)	HER-2+ MBC HER-2+ MBC HER-2+ MBC HER-2+ MBC HER-2+ MBC HER-2+ EBC	Tucatinib +T+P Tucatinib +ChT Tucatinib +T + ChT Tucatinib +T +ChT Tucatinib +T +ChT Neoadj tucatinib +T+P	Recruiting Recruiting Recruiting Recruiting Not yet recruiting Recruiting
Pyrotinib	NCT05346861 (III) NCT05255523 (II) NCT04605575 (II) NCT05429294 (II) NCT04246502 (II) NCT04254263 (III) NCT05841381 (III) NCT05861271 (II) NCT04659499 (II) NCT05880927 (II) NCT05834764 (II) NCT04929548 (II) NCT05659056 (II) NCT06000917 (II) NCT04917900 (II) NCT05430347 (II) NCT04900311 (II) NCT04290793 (II/III)	HER-2+ MBC HER-2+ MBC HER-2+ MBC HER-2+ MBC HER-2+ MBC HER-2+ EBC HER-2+ EBC HER-2+ EBC HER-2+ EBC HER-2+ EBC HER-2+ EBC HER-2+ EBC HER-2+ EBC HER-2+ EBC HER-2+ EBC HER-2+ EBC HER-2+ EBC HER-2+ EBC	Pyrotinib rechallenge Pyrotinib +T in ≥2L Pyrotinib +ChT Pyrotinib +T +ChT in 1L Pyrotinib +ChT in 1L Adj pyrotinib +T Adj pyrotinib +T +ChT Adj pyrotinib +ChT Adj pyrotinib +ChT Adj pyrotinib Adj pyrotinib Neoadj pyrotinib +T +P Neoadj pyrotinib +T +ChT Neoadj pyrotinib +T +ChT Neoadj pyrotinib +T +ChT Neoadj pyrotinib +T +ChT Neoadj pyrotinib +T +ChT Neoadj pyrotinib +ChT	Recruiting Not yet recruiting Recruiting Recruiting Not yet recruiting Recruiting Not yet recruiting Not yet recruiting Not yet recruiting Recruiting Recruiting Not yet recruiting Recruiting Recruiting Recruiting Not yet recruiting Not yet recruiting Not yet recruiting
T-DM1	NCT01853748 (II)	HER-2+ EBC	Adj T-DM1	Active, not recruiting
T-DXd	NCT04784715 (III) NCT05744375 (II) NCT04622319 (III) NCT05113251 (III) NCT05900206 (II) NCT05704829 (II) NCT05953168 (II)	HER-2+ MBC HER-2+ MBC HER-2+ EBC HER-2+ EBC HER-2+ EBC HER-2+ EBC HER-2-low MBC	T-DXd ±P in 1L T-DXd in 1L Adj T-DXd Neoadj T-DXd ±T + P +ChT Neoadj T-DXd Neoadj T-DXd T-DXd in 1L	Recruiting Recruiting Recruiting Recruiting Recruiting Not yet recruiting Not yet recruiting

Adj: Adjuvant; Neoadj: Neoadjuvant; ChT: Chemotherapy; EBC: Early breast cancer; ET: Endocrine therapy; MBC Metastatic breast cancer; P: Pertuzumab; T: Trastuzumab; 1L: First line; 2L: Second line.

## 4. Novel HER-2-Targeted Therapies 

Despite the great evolution in HER-2-targeted therapies and the outstanding improvement in HER-2-overexpressing breast cancer prognosis, there is still a long way to go in HER-2 breast cancer treatment. In the early setting, there is still a considerable number of relapses, and, in the metastatic setting, most cases eventually progress under HER-2-targeted therapy, due to either suboptimal tumor cell growth inhibition or mechanisms of resistance. For these reasons, there is continuous research and new drug development in this area, which is summarized below. 

### 4.1. Novel Monoclonal Antibodies

#### 4.1.1. MM-302

Status: Phase 1 clinical trials.

MM-302 is a liposome containing approximately 20.000 molecules of doxorubicin, and it is coated by 45 surface anti-HER-2 antibodies [61]. Preclinical studies demonstrated the efficacy of this drug in HER-2-overexpressing breast and gastric cancer and showed synergy with trastuzumab. A phase 2 clinical trial testing MM-302 in combination with trastuzumab in heavily pretreated patients showed, however, no clinical benefit, leading to the study’s early termination [62].

#### 4.1.2. Inetetamab

Status: Phase 2 clinical trials.

Inetetamab, also called cipterbin, is a novel anti-HER-2 mAb directed against subdomain IV of HER-2, with a slightly different aminoacid sequence in the Fc portion. Preclinical studies demonstrated efficacy against HER-2-overexpressing tumors [63,64]. A recently published retrospective study reported that inetetamab combined with vinorelbine and pyrotinib was effective in heavily pretreated HER-2-positive breast cancer patients [65].

An ongoing phase 2 trial is evaluating the role of inetetamab in the neoadjuvant setting, in combination with pertuzumab and chemotherapy. Five phase 2 studies are testing the combination of inetetamab with pyrotinib and chemotherapy in the metastatic setting, in first or subsequent lines (see Table 5).

### 4.2. Bispecific HER-2-Targeted Antibodies

Bispecific antibodies were developed with the aim of targeting different epitopes or domains in the HER-2 receptor, optimizing its blockade, or even targeting a different receptor besides HER-2, such as HER-3. This receptor plays an essential role in HER-2/HER-3 dimerization and downstream signaling of the PI3K pathway, one of the known mechanisms of resistance to HER-2-targeted therapy.

#### 4.2.1. Zanidatamab

Status: Phase 2 clinical trials.

Zanidatamab (ZW25) is a humanized IgG1 mAb that simultaneously targets subdomains II and IV of the HER-2 extracellular component. Each HER-2 receptor can be targeted by two molecules of this drug, which enhances HER-2 internalization and downregulation [66]. 

Zanidatamab has proven its efficacy in second-line treatment of HER-2-positive MBC in two phase 1 studies, whether in monotherapy [67] or in combination with chemotherapy [68]. There are currently ongoing clinical trials testing zanidatamab’s efficacy in combination with ADCs (NCT05027139), with CDK4/6 inhibitors (NCT04224272) and in monotherapy in the neoadjuvant setting (NCT05035836).

#### 4.2.2. MBS301

Status: Phase 1 clinical trials.

MBS301 is a bispecific IgG1 antibody targeting extracellular subdomains II and IV. It is glycol-engineered from trastuzumab and pertuzumab, consisting of one-half of each of these mAbs. In preclinical studies, this drug showed greater efficacy when compared to the trastuzumab and pertuzumab association, mostly due to an increased ADCC stimulation [69].

A phase 1 clinical trial is still ongoing in China (NCT03842085) and no preliminary data are yet available.

#### 4.2.3. Anbenitamab

Status: Phase 2 clinical trials.

Anbenitamab, also called KN026, is a bispecific antibody (bsAb) derived from trastuzumab and pertuzumab which showed efficacy in tumor cells resistant to these two drugs [70]. Phase 1 clinical trials showed a similar efficacy to the trastuzumab and pertuzumab combination even in heavily pretreated patients [71]. 

Preliminary data from two recently concluded phase 1 studies in HER-2-overexpressing breast and gastric cancer patients revealed promising antitumoral activity with good tolerability [71,72]. Ongoing clinical trials are evaluating KN026’s efficacy in the neoadjuvant treatment of breast cancer and in the metastatic setting in combination with chemotherapy (see Table 5).

#### 4.2.4. Zenocutuzumab 

Status: Phase 1 clinical trials.

Zenocutuzumab (MCLA-128) is another IgG1 bsAb, but instead of targeting different subdomains of HER-2, it targets subdomain I of HER-2 and the ligand-binding site of HER-3, thereby impeding HER-2-HER-3 dimerization and downstream signaling. Besides blocking the HER-2 signaling pathway, zenocutuzumab also strongly stimulates ADCC [73]. 

Zenocutuzumab demonstrated efficacy in combination with trastuzumab and vinorelbine in the third line and beyond in HER-2-positive MBC [74]. A recently published phase 1 study testing this drug’s efficacy in patients with NGR1 fusion demonstrated robust and durable efficacy with a good tolerability profile [75]. 

There is currently one ongoing phase 2 trial testing a combination with chemotherapy, trastuzumab or endocrine therapy in HER-2-positive or HER-2-low/hormonal receptor-positive breast cancer patients (NCT03321981).

#### 4.2.5. HER2(Per)-S-Fab

Status: Preclinical studies.

HER2(Per)-S-Fab is a bispecific antibody composed of pertuzumab antigen-binding fraction (Fab), which binds the HER-2 subdomain II and an anti-CD16/FcγRIIIa antibody [76]. The anti-CD16/FcγRIIIa portion exerts an immune stimulatory effect, by recruiting natural killer (NK) cells, which express CD16, to the tumoral microenvironment (TME) and promoting its cytotoxic effects against target HER-2-positive cells [76]. There are no ongoing clinical trials with this drug.

#### 4.2.6. HER2-2XCD16 

Status: Preclinical studies.

HER2-2XCD16 is a tribody which integrates an anti-HER-2 single chain fused with IFN- γ, leading to IFN-dependent cell death, even in tumors resistant to HER-2-blockage [77]. Besides an antitumoral effect in HER-2-overexpressing breast cancer cells, this molecule demonstrated an ability to interfere with the TME, switching it to an antitumoral environment [78].

#### 4.2.7. Discontinued Bispecific Antibodies 

There are three other bispecific antibodies that showed an antitumoral effect in HER-2-positive breast cancer but were, meanwhile, discontinued, by decision of the sponsor.

MM-111 targets both the extracellular component of HER-2 and the heregulin-binding site of HER-3, inhibiting both HER-2 and PI3K pathways [79]. Preclinical studies showed efficacy against HER-2-overexpressing breast cancer cells, and when combined with trastuzumab, it demonstrated more efficacy than trastuzumab with pertuzumab [79]. 

Ertumaxomab targets HER-2 and CD3 and has a trimodal action through the recognition of not only tumor cells, but also stromal cells and T-cells [80,81]. Preclinical studies demonstrated a moderate antitumoral effect in both HER-2-overexpressing and HER-2-low breast cancer, with inferior efficacy when compared with standard of care [81]. Phase 2 trials of ertumaxomab were early-terminated.

ISB1302 or GBR1302 also targets CD3 in T-cells and directs them to HER-2-overexpressing tumor cells, where they bind HER-2. Phase 1 studies confirmed activation of a T-cell-mediated immune response [82], but no further studies were conducted.

### 4.3. Novel Tyrosine Kinase Inhibitors (TKIs)

#### 4.3.1. Poziotinib

Status: Phase 2 clinical trials.

Poziotinib is an irreversible pan-HER TKI that inhibits EGFR, HER-2 and HER-4. Preclinical studies revealed an antitumoral effect in HER-2-overexpressing or HER-2-mutant breast, lung and gastric cancer lines [83], and demonstrated that poziotinib upregulates HER-2 expression in the cell membrane and potentiates the antitumoral effect of other anti-HER-2 therapies, such as T-DM1 [84]. Phase 1 studies confirmed the efficacy of poziotinib in the treatment of HER-2-positive tumors [85]. A phase 2 trial tested poziotinib’s efficacy in the treatment of HER-2-overexpressing metastatic breast cancer patients in the third line and beyond, showing meaningful antitumoral efficacy with overall survival benefits and a toxicity profile similar to other TKIs [86,87]. A phase 1b (NCT03429101) and other phase 2 studies were recently conducted in metastatic breast cancer patients, in monotherapy or association with T-DM1, both in HER-2-positive (NCT02659514; NCT02418689) and HER-2-mutant patients (NCT02544997), and preliminary data is still awaited (see Table 6). There are no clinical trials currently ongoing.

#### 4.3.2. DZD1516

Status: Phase 1 clinical trials.

DZD1516 is a reversible HER-2-specific TKI with full blood–brain barrier penetration [88]. In preclinical studies, it revealed an antitumoral effect in HER-2-positive tumors [88]. A phase 1 study from the same author revealed good disease control in heavily pretreated HER-2-positive breast cancer patients with brain metastases [88]. Another phase 1 study is currently ongoing (NCT04509596) and preliminary data confirms good intracranial disease control with a favorable toxicity profile [89]. 

#### 4.3.3. Discontinued TKIs

Epertinib, also called S-222611, is a reversible pan-HER TKI. In preclinical studies, it revealed a superior antitumoral effect in HER-2-overexpressing tumors when compared with lapatinib and a more favorable toxicity profile when compared with irreversible TKIs [90]. Phase 1 clinical trials confirmed an antitumoral effect and good tolerability in HER-2-overexpressing breast, upper digestive, head and neck and renal tumors in monotherapy [91], and in combination with trastuzumab with or without capecitabine [92]. Despite these encouraging results, no further trials were performed.

BDTX-189 is an irreversible EGFR/HER-2 TKI. Preliminary data from a recently concluded phase 1 trial, MasterKey-01, revealed an antitumoral effect in patients with HER-2/HER-3-mutated and HER-2-amplified solid tumors [93]. Despite the promising results, the drug’s sponsor decided to discontinue further research. 

### 4.4. Novel Antibody–Drug Conjugates (ADCs)

Despite the great efficacy of currently commercialized ADCs for HER-2-overexpressing breast cancer, mechanisms of resistance to T-DM1 and T-DXd have already been identified, and the development of further ADCs, with a greater therapeutic index and less toxicity, is needed, justifying continued research in this area.

#### 4.4.1. Trastuzumab Duocarmycin (SYD985)

Status: Phase 3 clinical trials.

Trastuzumab duocarmycin is an IgG1 ADC composed of trastuzumab seco-duocarmycin-hydroxybenzamide-azaindole (seco-DUBA), a prodrug of DUBA, a DNA alkylator cytotoxic drug, in a drug-to-antibody ratio of 2.8:1, lower than both T-DM1 and T-DXd. After binding to HER-2 and being internalized, the cleavable linker binding both components of the ADC is cleaved in the lysosome, and the active drug DUBA is released in the cytoplasm [94]. SYD985 has considerable bystander killing effect for two reasons—DUBA itself is membrane-permeable after being released from the lysosome, and SYD985 cleaving proteases, like cathepsin B, are usually produced by tumor cells, meaning they are present in the extracellular milieu, thereby maximizing this ADC’s effect in surrounding tumor cells [94]. 

Preclinical studies demonstrated SYD985’s efficacy in HER-2-overexpressing breast cancer cells, with similar HER-2 binding, HER2-mediated internalization and ADCC stimulation when compared with T-DM1 [95]. Phase 1 studies confirmed efficacy not only in HER-2-overexpressing but also in HER-2-low metastatic breast cancer, with a greater efficacy in the HER-2-low subset of patients when compared with T-DM1, probably due its potent bystander killing effect [94]. The results of the phase 3 clinical trial TULIP are still to be published but preliminary data reveal a significant improvement in progression-free survival with no significant increase in overall response rate or overall survival, when compared with physician’s choice in third line and beyond HER-2-overexpressing metastatic breast cancer [96]. 

SYD985 is currently being tested along with other targeted therapies in the neoadjuvant setting (NCT01042379—see Table 7).

#### 4.4.2. Trastuzumab Rezetecan

Status: Phase 2/3 clinical trials.

Trastuzumab rezetecan, also called SHR-A1811, is an ADC composed of trastuzumab and SHR9265, a novel topoisomerase I inhibitor derived from exatecan with a better liposolubility and cellular permeability. The two compounds are bonded by a stable and cleavable linker, in a drug-to-antibody ratio of 5.7:1 [97]. In preclinical studies, SHR-A1811 showed HER2-dependent antitumoral activity in breast and gastric cancer cell lines, resulting in a dramatic and sustained tumoral cell growth inhibition. It was also shown to have a significant bystander effect [97].

A recently published phase 1 study in HER2-expressing or -mutated solid tumors revealed that SHR-A1811 had a favorable antitumor effect with a manageable toxicity profile [98].

There are currently several ongoing trials testing SHR-A1811 in monotherapy or in combination with other agents—one phase 3 trial in HER-2-overexpressing metastatic breast cancer (NCT06057610), three in HER-2-low metastatic breast cancer (NCT05814354; NCT05845138; NCT05792410) and one in the neoadjuvant setting (NCT05635487).

#### 4.4.3. ARX788

Status: Phase 2/3 clinical trials.

ARX788 is an ADC composed of an anti-HER-2 mAb conjugated with monomethyl auristatin F (MMAF), a highly potent synthetic auristatin derivative that acts as a microtubule inhibitor. These two components are connected by a non-cleavable linker, as in T-DM1, with a unique site-specific conjugation technology, unlike other ADCs, in which cytotoxic payloads are randomly assembled to antibodies. This ADC’s drug-to-antibody ratio is 1.9:1, lower than existent ADCs [99]. 

Preclinical studies revealed efficacy in HER-2-overexpressing and HER-2-low breast tumor cells and even in cells resistant to T-DM1. It also showed a stronger antitumoral activity when compared with T-DM1 [99,100]. A phase 1 clinical trial confirmed its efficacy in HER-2-overexpressing metastatic breast cancer patients who progressed on previous HER-2-targeted therapies, with a manageable safety profile [101].

There are currently ongoing one phase 2 trial in the metastatic setting [(NCT04829604), and three phase 2/3 studies in the neoadjuvant setting, two in combination with the TKI pyrotinib (NCT04983121; NCT05426486) and one in monotherapy in a previously mentioned multi-arm study (NCT01042379).

#### 4.4.4. Disitamab Vedotin (RC48)

Status: Phase 2/3 clinical trials; approved in China, not approved by FDA or EMA.

Disitamab vedotin is an ADC that combines hertuzumab (a new anti-HER-2 mAb that targets HER-2 subdomain IV in a different epitope than trastuzumab) with monomethylauristatin E (MMAE), also known as vedotin, an inhibitor of tubuline polymerization during cell division [102,103]. This ADC integrates a cleavable linker and has an estimated drug-to-antibody ratio of 4:1.

In preclinical studies, RC48 demonstrated greater efficacy than trastuzumab and lapatinib against HER-2-overexpressing breast cancer cells and showed a greater antitumoral effect than T-DM1 in trastuzumab- and lapatinib-resistant tumor cells [102]. Ongoing phase 1 studies are assessing RC48’s efficacy in both HER-2-overexpressing and HER-2-low breast cancer (NCT02881138; NCT03052634). 

There are several ongoing phase 2/3 trials in China, testing the effect of disitamab vedotin in different subsets of breast cancer: two studies in HER-2-positive metastatic breast cancer (NCT05331326; NCT03500380); three studies in HER-2-low metastatic breast cancer (NCT04400695; NCT05904964; NCT06105008); and finally, two studies in the neoadjuvant setting of HER-2-positive breast cancer (NCT05134519; NCT05726175) (see Table 7).

Disitamab vedotin received approval in China in June 2021 for treatment of HER-2-positive locally advanced/metastatic breast cancer beyond the third line [104].

#### 4.4.5. Zanidatamab Zovodotin (ZW49)

Status: Phase 1 clinical trials.

Zanidatamab zovodotin is an ADC that combines zanidatamab, a bsAb targeting domains II and IV of the HER-2 extracellular component, and N-acyl sulfonamide auristatin, also called zovodotin, an auristatin that inhibits microtubule polymerization during cell division and that is associated with a more favorable toxicity profile than other auristatins such as MMAE and MMAF. It is bonded by a cleavable linker and has a drug-to-antibody ratio of 2:1 [105]. 

A preclinical study documented the antitumoral activity of ZW49 against HER-2-overexpressing and HER-2-low breast cancer cells and registered an increased binding and internalization of this ADC when compared with trastuzumab-based ADCs [105]. A phase 1 study testing ZW49 in HER-2-overexpressing solid tumors has been recently completed (NCT03821233). Preliminary data confirmed its efficacy in monotherapy, with a manageable toxicity profile in heavily pretreated HER-2-positive solid tumors [106].

#### 4.4.6. MRG002

Status: Phase 2/3 clinical trials.

MRG002 is an ADC composed of a recombinant humanized anti-HER-2 mAb similar to trastuzumab but with less ADCC stimulation, called MAB802, and MMAE, an auristatin previously mentioned. These compounds are connected by a protease-cleavable linker and the ADC’s estimated drug-to-antibody ratio is 3.8:1 [107]. 

Preclinical studies showed an antitumoral effect in HER-2-overexpressing breast cancer cells and greater efficacy when compared with both trastuzumab and T-DM1, with an antitumoral effect in T-DM1-resistant cancer cells [107].

There are currently three phase 2/3 clinical trials ongoing in China, in the metastatic setting for HER-2-overexpressing (NCT05263869; NCT04924699) and HER-2-low (NCT04742153) breast cancer (see Table 7).

#### 4.4.7. A166

Status: Preclinical studies.

A166 is a trastuzumab-based ADC that carries a duostatin-5 payload, a cytotoxic that acts by inhibiting microtubules. Duostatin-5 binds to trastuzumab by a protease-cleavable linker through a site-specific conjugation technology, in a drug-to-antibody ratio of 2:1. 

Phase 1 studies revealed promising the efficacy and tolerable toxicity of A166 in HER-2-positive advanced solid tumors refractory to standard HER-2-targeted therapies [108]. There is another phase 1 trial currently ongoing in China in HER-2-overexpressing solid tumors (NCT05311397), but there are still no studies specifically in a breast cancer setting.

#### 4.4.8. ALT-P7

Status: Phase 1 clinical trials.

ALT-P7 is another ADC composed by a trastuzumab biosimilar, HM2, and MMAE, a microtubule inhibitor, both connected by a cleavable linker in a ratio of 2:1 [109]. A phase 1 trial including HER-2 positive breast cancer patients that progressed under trastuzumab-based therapy was recently conducted (NCT03281824) and preliminary data reveals a favorable tolerability profile and indicates that there will probably be a phase 2 study [110].

#### 4.4.9. XMT-1522

Status: Phase 1 clinical trials.

XMT-1522 is an ADC composed of an anti-HER-2 mAb, HT-19, which binds subdomain IV of HER-2 in a different epitope from trastuzumab, combined with a microtubule-inhibiting auristatin derivative, AF-HPA, in a proportion of 12:1. This ADC has this abnormally high drug-to-antibody ratio due to a biodegradable polymer-based conjugation platform that enables these high ratios without plasmatic aggregation or interference with the drug pharmacokinetics [109,111].

Preclinical studies demonstrated XMT-1522 to be effective against HER-2-overexpressing breast cancer cells, even in the presence of T-DM1 resistance, and with an approximately 100 times higher potency than T-DM1 [112]. It also showed a synergistic effect with trastuzumab and pertuzumab [113] and revealed an antitumoral effect in HER-2-low breast cancer cells [114]. A phase 1 clinical trial testing this ADC in the metastatic HER-2-overexpressing breast cancer setting (NCT02952729) was recently concluded, and preliminary data showed a good tolerability and antitumor activity [115].

#### 4.4.10. Discontinued ADCs 

PF-06804103 is composed of a trastuzumab-derived antibody associated with Aur0101, a potent microtubule polymerization inhibitor, with site-specific conjugation technology, in a proportion of 4:1 [116]. Preclinical studies showed antitumor activity against both HER-2-positive and HER-2-low breast cancer, showing greater response rates and more durable complete responses when compared with T-DM1 [117]. A recently published phase 1 trial confirmed antitumor activity in HER-2-positive and HER-2-low breast and gastric cancers but revealed significant toxicity in almost half of the patients [118], leading to the drug’s discontinuation.

Alta-ADC is an ADC composed of pertuzumab and MMAE in a proportion of 2:1. It has a particularly low affinity for HER-2 at low pH, allowing its dissociation in the endosome, and subsequent pertuzumab release to bind a different HER-2 receptor in the cell membrane. This particularity allows Alta-ADC to be effective in lower doses, which warrants a more manageable toxicity profile [119]. Despite preclinical studies showing superior efficacy when compared with T-DM1 and also efficacy in HER-2-low tumors, no more studies were conducted. 

MEDI4276 is an ADC that combines a HER-2-bispecific antibody, which binds HER-2 in two different epitopes from trastuzumab and pertuzumab, and a novel microtubule inhibitor called AZ13599185, both connected by a cleavable linker, in a ratio of 4:1 [120,121]. Preclinical studies revealed a 10-fold more potent antitumoral effect than T-DM1 and also efficacy in T-DM1-resistant and HER-2-low tumor cells [121]. A phase 1/2 study conducted in 2016–2017, in heavily pretreated HER-2-overexpressing breast and gastric cancer patients, showed unacceptable toxicity [122], leading to its discontinuation. 

DHES0815A is composed of an anti-HER-2 mAb linked to the cytotoxic drug pyrrolo [2,1-c][1,4] benzodiazepine monoamide (PBD-MA), which creates crosslinks in DNA minor grooves, leading to DNA strand breaks [123]. A phase 1 study revealed a modest antitumoral effect with significant toxicity [124], which led to drug discontinuation in the setting of breast cancer.

### 4.5. Other Antibody Conjugates

#### 4.5.1. Immune-Stimulating Antibody Conjugates (ISAC)

BDC-1001 is not an ADC, as it includes no cytotoxic molecule, but is an immune-stimulating antibody conjugate, as it carries an immune-stimulating agent. It is composed of a trastuzumab biosimilar and a TLR-7/8 agonist that stimulates antigen-presenting cells to recognize and kill HER-2-overexpressing tumor cells, promoting a durable adaptive immune response. The two components are bound by a non-cleavable linker.

In preclinical studies, BDC-1001 showed an antitumoral effect through a solid immune response [125].

There are currently ongoing two phase 2 studies, one in HER-2-overexpressing metastatic breast cancer in combination with pertuzumab (NCT05954143), and one in HER-2-overexpressing solid tumors in combination with nivolumab (NCT04278144).

#### 4.5.2. Radionuclide–Antibody Conjugates

Thorium-227 conjugates (TTCs) are a combination of monoclonal antibodies with α-emitting radionuclides (thorium-227) that induce DNA double strand breaks, resulting in cell death, and they also have the capacity to stimulate an immunogenic response [126]. BAY2701439 is a TTC composed of an anti-HER-2, trastuzumab-derived antibody, conjugated with thorium-227. Preclinical studies demonstrated an antitumoral effect in HER-2-overexpressing tumor cells, including breast cancer cells, and in this set of patients, BAY2701439 also revealed efficacy in HER-2-low and T-DM1-resistant tumors [127]. A phase 1 clinical trial testing BAY2701439 in HER-2-overexpressing solid tumors was recently concluded, and preliminary data is awaited (NCT04147819).

### 4.6. Combinations of HER-2-Targeted and Other Agents

#### 4.6.1. Combination of HER-2-Targeted and Other Targeted Agents

HER-2-positive breast cancer is a heterogeneous disease, with tumors having different levels of hormonal receptor expression and a wide range of genomic alterations. Therefore, there is a growing interest in assessing the benefit of combining HER-2-targeted agents with other targeted therapies. Table 8 lists the main ongoing trials testing the combination of HER-2-targeted and other targeted agents.

#### 4.6.2. Combination of HER-2-Targeted Agents and Immunotherapy

Besides the direct inhibition of HER-2 signaling pathways through HER-2-targeted therapies, leading to cell cycle arrest and cell growth inhibition, there is also an important role for immune-mediated cell death in HER-2-overexpressing tumors. Ever since the development of the first HER-2-targeted drug, trastuzumab, the immune system proved to play an important role in tumor growth inhibition and apoptosis through ADCC and ADCP [17,18]. This cytotoxic mechanism is so important that its blockade by HLA class I (such as HLA-G)-mediated NK cells’ inhibition constitutes an important resistance mechanism to trastuzumab [128]. Besides ADCC and ADCP, trastuzumab was also shown to stimulate the production of TGFβ and IFN-γ by tumor cells and NK cells, which leads to the expression of PD-1 on NK cells [128].

These parallel mechanisms of action prove that immune stimulation may be of great importance in HER-2-positive tumor cells’ death, and it may enhance HER-2 inhibition and help overcome resistance mechanisms to HER-2-targeted agents. Immune stimulation in this context has the additional advantage of triggering immune memory against HER-2 cancer cells and providing a more durable and sustained antitumoral response [129].

This premise has led to the conceptualization of novel immune-stimulating drugs and to the evaluation of the benefit of several combinations of immune checkpoint inhibitors (ICI) with HER-2-targeted agents. A phase 1 study published in 2019 confirmed that pembrolizumab in association with trastuzumab had antitumor efficacy in previously trastuzumab-resistant patients with both HER-2 and PD-L1 positivity [130]. A recently published phase 3 trial showed, however, no benefit in adding an ICI, atezolizumab, to neoadjuvant chemotherapy and double HER-2 blockade, regarding the rate of pCR [131]. Table 9 summarizes the main ongoing clinical trials testing combinations of ICI with HER-2-targeted agents.

## 5. Other exploratory Therapies in HER-2-Positive Breast Cancer

### 5.1. PROTACs

Proteolysis-targeting chimeras (PROTACs) are stimulators of the natural ubiquitin-protease system. They are composed of two ligands connected by a linker—one ligand that binds the target protein and another ligand that binds a ubiquitin ligase. The binding of PROTACs to their targets induces ubiquitylation and degradation of the target protein by the ubiquitin-proteasome system, after which the PROTAC regenerates to bind other targets [132].

Ab-PROTAC 3 is a trastuzumab-PROTAC conjugate, connected by a cleavable linker, that induces ubiquitin-proteasome-mediated degradation of bromodomain-containing 4 (BDR4), a protein that plays an essential role in DNA replication during cell division. Preclinical studies showed that this conjugate can selectively target BDR4 in HER-2-positive tumor cells [133].

ORM-5029 is a pertuzumab–PROTAC conjugate that targets GSTP1, a protein important in protein synthesis in the ribosome and thus cell survival. Preclinical studies revealed a ten- to a thousand-fold greater potency than trastuzumab in suppressing HER-2-positive cancer cells, and it also showed an effect in HER-2-low cancer. There is a phase 1 ongoing trial testing ORM-5029 in HER-2-positive solid tumors (NCT05511844).

### 5.2. Cell Therapies

Cell-based immunotherapy is an innovative therapeutic modality, currently in use only in hemato-oncology, that uses genetically engineered immune cells from the patient to target and eliminate specific tags in tumor cells. After removing immune cells (T cells, NK cells or macrophages) from the patient’s tumoral microenvironment to modify their receptors accordingly, these cells are expanded ex vivo and reimplanted in the patient’s circulation to boost the patient’s antitumoral immune response. Despite success in liquid tumors, implementation of cell-based therapies in solid tumors has been full of obstacles, regarding the increased difficulty of accessing immune cells in the tumor, the triggered systemic inflammatory response syndrome (SIRS) and the side effects profile [134,135].

There are currently a few ongoing phase 1 clinical trials in HER-2-positive tumors, including with CAR-T cells (NCT04511871), CAR-NK cells (NCT05385705) and with CAR-macrophages (NCT04660929).

### 5.3. Cancer Vaccines

Another early and exploratory therapeutical modality is cancer vaccines. Cancer vaccines grant the advantages of having a very mild and manageable toxicity profile when compared with any other antitumoral treatment, even targeted HER-2 therapies, a very favourable posology, with no need for frequent administrations, and of providing a long-term and durable immune response specifically targeted against the tumor [136,137]. Cancer vaccines can be peptide-based, protein-based, cell-based, made of dendritic cells, virus-based or made of recombinant DNA.

The first vaccine to show benefits in cancer was SipuleucelT, which proved to be effective in prostate cancer [138]. Nelipepimut-S is a vaccine that combines E75, a peptide that mimics the extracellular domain of the HER-2 receptor, and granulocyte-macrophage colony-stimulating factor. Preclinical studies demonstrated E75 recognition by CD8+ lymphocytes and subsequent HER-2-positive cells death [139]. Phase 1 and 2 studies confirmed its safety and efficacy in triggering an immune response [140,141,142]. A phase 3 study failed, however, to show a disease-free survival benefit in HER-2-positive early breast cancer patients, leading to the study’s early termination [143]. A vaccine integrating a different peptide, GLSI-100, revealed an effective immune response in a phase 2b trial [144] and is currently being evaluated in a phase 3 trial (NCT05232916).

There are several other molecules undergoing evaluation in ongoing phase 1 and 2 trials (for example, NCT04418219; NCT03384914; NCT03387553; NCT05378464; NCT05325632).

## 6. Conclusions and Future Directions

Ever since the discovery and description of HER-2, the landscape of HER-2-overexpressing breast cancer has been constantly evolving. Trastuzumab was the first-ever targeted drug developed, more than a quarter of a century ago, bringing a shift of paradigm in the prognosis of this subset of patients, so much that it is still part of the standard of care. Since then, several other effective targeted agents have been developed and successfully integrated into clinical practice, contributing to a continuous improvement in the treatment of this subset of patients. Despite this constant improvement, drug resistance still occurs, and the disease still progresses under HER-2-targeted therapy, mostly due to suboptimal HER-2 inhibition or to mechanisms of resistance. It makes it essential to keep investing in the development of new drugs and better therapeutical options.

Thanks to a joint effort of basic scientists, clinical investigators and the pharmaceutical industry, continued research in this area has been bringing to light multiple and promising novel HER-2-targeted drugs, which optimize HER-2 blockage and overcome some of the resistance mechanisms identified so far. There is a special focus on the recently discovered class of antibody–drug conjugates, which allow scientists to combine HER-2 targeting with a tailor-delivered cytotoxic molecule with limited toxicity, and progress in this area keeps growing. The focus has not only been on HER-2-targeted drugs; other classes of drugs have also shown a benefit and synergistic effect with anti-HER-2 therapies, thereby improving treatment efficacy and diversifying future therapeutical weapons. Even immunotherapy and other forms of immune stimulation have been showing a significant benefit in this disease previously considered “cold” and non-immune-sensitive.

Despite many encouraging discoveries in the last few years, there is still a long way to go towards improving HER-2-positive breast cancer patients’ lives. There are currently 662 ongoing clinical trials, and many more promising to come up soon. Other flourishing therapeutical possibilities, still very exploratory, include single-cell sequencing, transcriptomics, proteomics, theragnostics and even modulating the microbiome. Advances in molecular diagnosis could help identify tumor-specific resistance mechanisms and surrogate biomarkers, allowing for early resistance detection and the tailoring of therapeutical options.

## Figures and Tables

**Table 5 cancers-16-00087-t005:** Summary of ongoing or recent clinical trials involving monoclonal antibodies (mAb) and bispecific antibodies (bsAb).

mAb/bsAb	Ongoing Clinical Trials (Phase)	Population	Treatment in Study	Status
MM-302	None	-	-	-
Inetetamab	NCT05749016 (II) NCT05823623 (II) NCT04681911 (II) NCT05823623 (II) NCT05621434 (II) NCT04963595 (II)	HER-2+ EBC HER-2+ MBC HER-2+ MBC HER-2+ MBC HER-2+ MBC HER-2+ MBC	Neoadj inetetamab +P +ChT Inetetamab +pyrotinib +ChT Inetetamab +pyrotinib +ChT Inetetamab +pyrotinib +ChT Inetetamab +pyrotinib +ChT in 1L Inetetamab +pyrotinib +ChT in 1L	Recruiting Recruiting Recruiting Recruiting Recruiting Not yet recruiting
Zanidatamab (ZW25)	NCT05027139 (I) NCT04224272 (II) NCT05027139 (Ib/II) NCT05035836 (II) NCT01042379 (II)	HER-2+ MBC HER-2+/HR+ MBC HER-2+ MBC HER-2+ EBC HER-2+ EBC	ZW25 ZW25 + palbociclib+fulvestrant ZW25 +ALX148 Neoadj ZW25 Neoadj ZW25	Concluded; unpublished Concluded; unpublished Recruiting Recruiting Recruiting
MBS301	NCT03842085 (I)	HER-2+ MBC	MBS301	Recruiting
Anbenitamab (KN026)	NCT03847168 (I) NCT04165993 (II) NCT04881929 (II)	HER-2+ MBC HER-2+/low MBC HER-2+ EBC	KN026 KN026 ±ChT Neoadj KN026 +ChT	Concluded; unpublished Active, not recruiting Recruiting
Zenocutuzumab (MCLA-128)	NCT02912949 (I) NCT03321981 (II)	HER-2+ HER-2+/HER-2-low/HR+ MBC	MCLA-128 MCLA-128 +T+ChT/+ET	Published Active, not recruiting
HER(Per)-S-Fab	None	-	-	-
HER2-2XCD16	None	-	-	-

Adj: Adjuvant; ChT: Chemotherapy; EBC: Early breast cancer; MBC: Metastatic breast cancer; Neoadj: Neoadjuvant; P: Pertuzumab; T: Trastuzumab.

**Table 6 cancers-16-00087-t006:** Summary of ongoing or recent clinical trials involving tyrosine kinase inhibitors (TKIs).

TKI	Target	Effect	Clinical Trials (Phase)	Population	Status
Poziotinib	Pan-HER (HER-1,2,4)	Irreversible	NCT02418689 (II) NCT02659514 (II) NCT03429101 (Ib) NCT02544997 (II)	HER2+ MBC HER+ MBC HER2+ MBC HER2m MBC	Published Completed; unpublished Completed; unpublished Completed; unpublished
DZD1516	HER-2	Reversible	NCT04509596 (I)	HER2+ MBC	Ongoing

MBC: Metastatic breast cancer; HER2m: HER-2-mutated; HER2+: HER-2 positive.

**Table 7 cancers-16-00087-t007:** Summary of antibody–drug conjugates (ADCs) and other antibody conjugates currently in study.

ADC/Antibody Conjugates	Payload	Antibody-to-Drug Ratio	Linker	Clinical Trial (Phase)	Setting	Status
Trastuzumab duocarmycin (SYD985)	Seco-DUBA	2.8:1	Cleavable	NCT03262935 (III) NCT01042379 (II)	HER-2+ MBC in ≥3L HER-2+ EBC (neoadj)	Closed; unpublished Recruiting
Trastuzumab rezetecan (SHR-A1811)	SHR9265	5.7:1	Cleavable	NCT06057610 (III) NCT05814354 (III) NCT05845138 (I/II) NCT05792410 (Ib/II) NCT05635487 (II)	HER-2+ MBC HER-2-low MBC HER-2-low MBC HER-2-low MBC HER-2+ EBC (neoadj)	Recruiting Recruiting Recruiting Recruiting Recruiting
ARX788	MMAF	1.9:1	Non- cleavable	NCT04829604 (II) NCT04983121 (II) NCT05426486 (II/III) NCT01042379 (II)	HER-2+ MBC HER-2+ EBC (neoadj) HER-2+ EBC (neoadj) HER-2+ EBC (neoadj)	Recruiting Recruiting Recruiting Recruiting
Disitamab vedotin (RC48)	MMAE	4:1	Cleavable	NCT03052634 (I/II) NCT05331326 (II) NCT03500380 (II/III) NCT04400695 (III) NCT05904964 (III) NCT06105008 (II) NCT05134519 (II) NCT05726175 (II)	HER-2+/low MBC HER-2+ MBC HER-2+ MBC HER-2-low MBC HR+/HER-2-low MBC HR+/HER-2-low MBC HER-2+ EBC (neoadj) HER-2+ EBC (neoadj)	Closed; unpublished Recruiting Recruiting Recruiting Recruiting Not yet recruiting Not yet recruiting Not yet recruiting
Zanidatamab zovodotin (ZW49)	Zovodotin	2:1	Cleavable	NCT03821233 (I)	HER-2+ solid tumors	Closed; unpublished
MRG002	MMAE	3.8:1	Cleavable	NCT05263869 (II) NCT04924699 (II/III) NCT04742153 (II)	HER-2+ MBC HER-2+ MBC HER-2-low MBC	Recruiting Recruiting Recruiting
A166	Duostatin-5	2:1	Cleavable	NCT03602079 (I/II) NCT05311397 (I)	HER-2+ solid tumors	Published Recruiting
ALT-P7	MMAE	2:1	Cleavable	NCT03281824 (I)	HER-2+ MBC	Closed; unpublished
XMT-1522	AF-HPA	12:1	Cleavable	NCT02952729 (I)	HER-2+ MBC	Closed; unpublished
BAY2701439	Thorium-227	Unknown	Unknown	NCT04147819 (I)	HER-2+ solid tumors	Closed; unpublished
BDC-1001	TLR-7/8 agonist	Unknown	Non- cleavable	NCT04278144 (II)	HER-2+ solid tumors Metastatic HER-2+ BC	Recruiting Recruiting

Adj: Adjuvant; EBC: Early breast cancer; MBC: Metastatic breast cancer; Neoadj: Neoadjuvant.

**Table 8 cancers-16-00087-t008:** Summary of ongoing clinical trials testing combinations of HER-2-targeted agents with other targeted agents.

Combination Agent	Clinical Trials (Phase)	Population	Treatment in Study	Status
CDK4/6 inhibitors	NCT05577442 (II) NCT05969184 (II) NCT03304080 (I/II) NCT02448420 (II) NCT03530696 (II) NCT03054363 (II) NCT05076695 (II) NCT05076695 (II) NCT04858516 (II) NCT03913234 (Ib/II) NCT05319873 (I/II) NCT02657343 (I/II)	HR+/HER-2+ MBC HR+/HER-2+ MBC HR+/HER-2+ MBC HR+/HER-2+ MBC HR+/HER-2+ MBC HR+/HER-2+ MBC HR+/HER-2+ MBC HR+/HER-2+ EBC HR+/HER-2+ MBC HR+/HER-2+ MBC HR+/HER-2+ MBC HR+/HER-2+ MBC	Dalpiciclib +pyrotinib +ET Palbociclib + T +P +ET Palbociclib +T +P +ET Palbociclib +T +ET Palbociclib +T-DM1 Palbociclib +tucatinib +ET Palbociclib +pyrotinib +T +F Neoadj palbociclib +pyrotinib +T +F Neoadj palbociclib +pyrotinib +T +ET Ribociclib +T +ET Ribociclib +tucatinib +T Ribociclib +T/T-DM1	Not yet recruiting Recruiting Active, not recruiting Active, not recruiting Completed; unpublished Completed; unpublished Recruiting Recruiting Not yet recruiting Recruiting Recruiting Completed; unpublished
PARP inhibitors	NCT03368729 (I/II)	HER-2+ mBRCA MBC	Niraparib +T	Recruiting
PIK3CA inhibitors	NCT03765983 (II) NCT04108858 (I/II) NCT02705859 (I) NCT05230810 (I/II) NCT04208178 (III) NCT05063786 (III)	HER-2+ mPIK3CA MBC HER-2+ mPIK3CA MBC HER-2+ mPIK3CA MBC HER-2+ mPIK3CA MBC HER-2+ mPIK3CA MBC HER-2+ mPIK3CA MBC	GDC-0084 +T Copanlisib +T +P Copanlisib +T Alpelisib +tucatinib Alpelisib +P +T in 1L Alpelisib +T ±F	Recruiting Recruiting Completed Recruiting Active, not recruiting Recruiting
ATK inhibitors	NCT04253561 (I)	HER-2+ mAKT MBC	Ipatasertib +T +P	Recruiting

Adj: Adjuvant; Neoadj: Neoadjuvant; ChT: Chemotherapy; EBC: Early breast cancer; ET: Endocrine therapy; F: fulvestrant; HR: Hormone receptors; MBC Metastatic breast cancer; mAKT: AKT mutated; mBRCA: BRCA mutated; mPIK3CA: PIK3CA mutated; P: Pertuzumab; T: Trastuzumab.

**Table 9 cancers-16-00087-t009:** Summary of ongoing clinical trials evaluating the combination of HER-2-targeted agents with immune-checkpoint inhibitors (ICI).

Combination Agent	Clinical Trials (Phase)	Population	Treatment in Study	Status
Atezolizumab	NCT03199885 (III) NCT03125928 (II) NCT04759248 (II) NCT04740918 (III) NCT04873362 (III) NCT03595592 (III)	HER-2+ MBC HER-2+ MBC HER-2+ MBC HER-2+ PD-L1+ MBC HER-2+ EBC HER-2+ EBC	Atezolizumab +T +P +ChT Atezolizumab +T +P +ChT Atezolizumab +T +ChT Atezolizumab +T-DM1 Adj atezolizumab +T-DM1 Neoadj atezolizumab +T +P	Active, not recruiting Active, not recruiting Recruiting Recruiting Recruiting Active, not recruiting
Durvalumab	NCT03820141 (II) NCT05795101 (II) NCT02649686 (I)	HER-2+ MBC HER-2+/low MBC HER-2+ MBC	Durvalumab +T +P in 1L Durvalumab +T-DXd Durvalumab +T	Recruiting Recruiting Completed
Pembrolizumab	NCT04789096 (II) NCT03032107 (Ib) NCT03988036 (II) NCT03747120 (II)	HER-2+ MBC HER-2+ MBC HER-2+ EBC HER-2+ EBC	Pembrolizumab +tucatinib +T ±ChT Pembrolizumab +T-DM1 Neoadj pembrolizumab +T +P Neoadj pembrolizumab +T +P +ChT	Recruiting Active, not recruiting Completed Recruiting

Adj: Adjuvant; Neoadj: Neoadjuvant; ChT: Chemotherapy; EBC: Early breast cancer; ET: Endocrine therapy; MBC Metastatic breast cancer; P: Pertuzumab; T: Trastuzumab; 1L: First line.

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
