# Peer review of "Novel HER-2 Targeted Therapies in Breast Cancer"

_cancers, 2023, doi:10.3390/cancers16010087_

Round 1

Reviewer 1 Report

Comments and Suggestions for Authors

The abstract is well written, and summarised the aim of the review paper to summarise the newly approved and developing HER2-targeted therapeutic options. The manuscript itself is well structured into approved and developing mAbs, TKIs, ADC and other approaches. The background of HER2 testing, targeting and the HER2 family is adequately introduced. Due to the well structured, the review read logically.

I have some minor comments below.

Line 26- Breast cancer is not the most common type of cancer in both sexes. It is the most common type of cancer in females but prostate cancer is most prevalent in males.

Line 43 – references to trastuzumab resistance is warranted here considering the strong statement.

Line 44 – This line is not right as after trastuzumab/pertuzumab, HER2 BCa has become a very treatable disease and outcomes have dramatically improved, even more so if it is HER2+/HR+ localised or regional. This line directly contradicts Line 31-32. I suggest emphasizing the continued development of HER2 therapeutics in the metastatic or refractory disease setting.

Throughout the manuscript – please standardise using either HER2 or HER-2.

Line 68 – what important role are you referring to regarding HER2-HER3 dimerisation?

There were numerous mention of the subdomains of HER2 in each of the drugs. Therefore, you need to elaborate HER2 domains in the HER receptor section. Particularly the ones which have clinical significance eg. Dimerization domain

Line 72 – what is the difference between cell growth and cell proliferation?

Line 213 – what is the mechanism of action of Tucatinib

Line 235 – what is the mechanism of action of Pyrotinib  and what is the status of Pyrotinib in the US?

Line 280-283 – This information is important but poorly written.

Line 322-324 – This is not clear. Is MM-302 is a liposome containing 45 mAb and 20 moles of DOX? What do you mean by associated?

4.2.5 – elaborate on the mechanism of action for HER2(Per)-S-Fab, - why target CD16

Line 442 – reference for this is important due to potential to treat HER2+BC pt with brain mets.

Line 610-612 – This is not clear, please rephrase.

Line 648 – references for the studies

Line656-657 This is not clear, please rephrase.

References for Lines 148, 171, 183 are essential.

HER-targeting can often induce adverse events such as diarrhoea being most common followed by heart pathologies. Further, potential adverse reactions play an important factor in choosing the treatment. Where notable, elaborate on the details of tolerability.

Comments on the Quality of English Language

The overall quality of English is fine. Though not critical, there are parts where the authors have great information but the meaning is lost/unclear due to poor wording. I would suggest getting it proofread to improve the quality of the manuscript.

Reviewer 2 Report

Comments and Suggestions for Authors

vThe authors present a thorough and exhaustive review of HER2-positive breast cancer therapies such as antibodies, TKIs and ADCs. In this section, they clearly focus on the approved therapies and briefly highlight the results of these trials. In later sections, novel, as yet unapproved anti-HER2 therapies such as bispecific antibodies or vaccines and also combinations with immunotherapies are presented.
All in all, this comprehensive review provides an overview of the state of the art in HER2-targeted therapies. However, for the sake of completeness, I have a few suggestions and comments.
I suggest including the negative neoadjuvant phase III trial of the combination of dual blockade with atezolizumab (PMID: 35763704  ). With regard to vaccines, the negative study on nelipepimut-S should also be briefly mentioned (PMID: 31036542  ) and the fact that a phase III study with a vaccination against HER2 in the adjuvant situation is ongoing (NCT05232916).

Reviewer 3 Report

Comments and Suggestions for Authors

The review summarizes the HER2-targeted therapies in breast cancer patients. HER2-positive breast cancer is diagnosed in about 20% of patients, it is more aggressive and bezforemnego the introduction of trastuzumab to treatment there was no targeted therapy for there patients. The authors describe the therapies in use - 8 drugs and much more in clinical trials. The information about the preclinical studies is very interesting especially about the cel therapies and cancer vaccine.
